# Does Tumor Volume Have a Prognostic Role in Oropharyngeal Squamous Cell Carcinoma? A Systematic Review and Meta-Analysis

**DOI:** 10.3390/cancers14102465

**Published:** 2022-05-17

**Authors:** Elena Russo, Remo Accorona, Oreste Iocca, Andrea Costantino, Luca Malvezzi, Fabio Ferreli, Ciro Franzese, Marta Scorsetti, Pasquale Capaccio, Giuseppe Mercante, Giuseppe Spriano, Armando De Virgilio

**Affiliations:** 1Otorhinolaryngology Unit, IRCCS Humanitas Research Hospital, Via Manzoni 56, 20089 Rozzano, Italy; andrea.costantino@humanitas.it (A.C.); luca.malvezzi@humanitas.it (L.M.); fabio.ferreli@humanitas.it (F.F.); giuseppe.mercante@humanitas.it (G.M.); giseppe.spriano@hunimed.eu (G.S.); armando.de_virgilio@humanitas.it (A.D.V.); 2Department of Biomedical Sciences, Humanitas University, Via Rita Levi Montalcini 4, 20090 Pieve Emanuele, Italy; ciro.franzese@hunimed.eu (C.F.); marta.scorsetti@hunimed.eu (M.S.); 3Department of Otorhinolaryngology—Head and Neck Surgery, Fondazione IRCCS Ca’ Granda Ospedale Maggiore Policlinico, 20122 Milan, Italy; remo.accorona@policlinico.mi.it (R.A.); pasquale.capaccio@unimi.it (P.C.); 4Division of Maxillofacial Surgery, Surgical Science Department, University of Torino, 10124 Torino, Italy; oi243@nyu.edu; 5Radiotherapy Unit, IRCCS Humanitas Research Hospital, Via Manzoni 56, 20089 Rozzano, Italy

**Keywords:** tumor volume, squamous cell carcinoma, oropharyngeal cancer, locoregional control, survival

## Abstract

**Simple Summary:**

Increasing evidence supports the role of tumor volume as a prognostic marker in head and neck cancer (HNC), specifically, in the glottis, supraglottis, hypopharynx and nasopharynx treated by primary radiotherapy. However, studies on oropharyngeal carcinomas have yielded mixed results. We performed a systematic review and meta-analysis to assess the prognostic value of tumor volume in oropharyngeal squamous cell carcinoma (OPSCC). In particular, we calculated the hazard ratio (HR) of overall survival (OS), disease-free survival (DFS) and locoregional control (LRC) of high values for primary and nodal tumor volume (pTV and nTV, respectively), compared to those of low values. Our findings based on 1417 patients with OPSCC showed that pTV and nTV are not predictors of OS, and they may not be used as prognostic factors in OPSCCs. Moreover, the difference in terms of DFS and LRC was too small to appear clinically relevant.

**Abstract:**

The aim of this study was to assess the prognostic value of tumor volume in oropharyngeal squamous cell carcinoma (OPSCC). The study was performed according to the PRISMA guidelines. A total of 1417 patients with a median age of 59.3 years (IQR 57.5–60) were included. The combined Hazard Ratios (HRs) for overall survival (OS) were 1.02 (95% CI, 0.99–1.05; *p* = 0.21) for primary tumor volume (pTV) and 1.01 (95% CI, 1.00–1.02; *p* = 0.15) for nodal tumor volume (nTV). Regarding locoregional control (LRC), the pooled HRs were 1.07 (95% CI, 0.99–1.17; *p* = 0.10) for pTV and 1.02 (95% CI, 1.01–1.03; *p* < 0.05) for nTV. Finally, the pooled HRs for disease-free survival (DFS) were 1.01 (95% CI, 1.00–1.03; *p* < 0.05) for pTV and 1.02 (95% CI, 1.01–1.03; *p* < 0.05) for nTV. In conclusion, pTV and nTV seem not to behave as reliable prognostic factors in OPSCC.

## 1. Introduction

Over the past three decades, the incidence of oropharyngeal squamous cell carcinoma (OPSCC) has increased, mirroring the increasing prevalence of Human Papilloma Virus (HPV) infection [1,2,3]. HPV-positive OPSCC has been shown to have a distinct biological behavior, with a favorable outcome and a better response to (chemo)radiotherapy compared to HPV-negative tumors [4,5,6]. However, there is still an unexplained variation in OPSCC prognosis and treatment outcome irrespective of HPV status.

The tumor, node, metastasis (TNM) classification is the most commonly used staging system to describe head and neck cancer (HNC). It is based on tumor dimension and extension with metric and anatomic criteria. Efforts have been made to improve the current staging system by investigating other factors that may have a prognostic impact. Increasing evidence supports the role of tumor volume as a prognostic marker in HNC. Several studies have shown that tumor volume may be superior to the TNM staging as a prognostic indicator among patients with carcinomas of the glottis, supraglottis, hypopharynx and nasopharynx treated by primary radiotherapy [7,8,9,10,11,12,13]. However, studies on oropharyngeal carcinomas have yielded mixed results. Several earlier studies observed no correlation between tumor volume and locoregional control in OPSCC [8,14,15,16]. In contrast, other studies showed that higher volume may correlate with an increased risk of recurrence and poorer survival [17,18,19,20,21]. In addition, the best volume of cut-off values able to predict the oncological outcome remains undefined.

The aim of this systematic review and meta-analysis was to assess the prognostic value of tumor volume in OPSCC. In particular, we systematized the currently available data to better define if volumetric staging together with TNM staging may be reliable in predicting the outcome of OPSCC.

## 2. Materials and Methods

The study was conducted with reference to the Preferred Reporting Items for Systematic Reviews and Meta-Analyses (PRISMA) statement [22]. Neither ethics approval nor informed consent were required for this review of previously published studies. The present study was registered in “Open Science” in 2022 (OSF DOI:10.17605/OSF.IO/YF7VQ).

### 2.1. Eligibility Criteria

Inclusion criteria comprised full text available studies with adult patients (>18 years of age) suffering from OPSCC with no distant metastasis, treated with surgery alone, surgery and adjuvant (chemo)radiotherapy, radiotherapy alone, or concurrent chemoradiotherapy. Before treatment, patients had to have undergone a CT or an MRI with the following volumetric parameters measurements: primary tumor volume (pTV), nodal tumor volume (nTV) and/or total tumor volume (tTV). Studies were required to provide survival analysis stratifying pTV, nTV and/or tTV in terms of overall survival (OS), disease-free survival (DFS) and/or locoregional control (LRC).

Exclusion criteria were as follows: non-English language; full text not available; insufficient reported data; cohorts composed not exclusively of patients with OPSCC; not extractable data; article type was either review, case report, conference abstract, letter to the editor, or book chapter.

### 2.2. Data Source and Study Searching

The PubMed/MEDLINE, Cochrane Library, and Google Scholar databases were searched for pertinent articles. Relevant keywords, phrases, and MeSH terms were adjusted to fit the specific requirements for each of the individual databases. An example of a search strategy was the one used for PubMed/MEDLINE: “prognosis” and “tumor volume” and “head and neck neoplasms” or “oropharyngeal neoplasms”. Then, a cross-referencing from the included articles was performed to ensure the retrieval of all possible studies. The last search was conducted on 16 November 2020.

### 2.3. Data Collection Process

Literature searches were conducted independently by two investigators (E.R. and A.D.V.). Initially, all papers were screened for relevance by title and abstract. Then, the authors independently assessed each full text that was considered relevant. Any conflict between reviewers was solved by consensus. Data extraction from the included studies was conducted independently by two reviewers (E.R. and A.D.V.), and the following information was recorded: first author, year of publication, study design, number of patients, patient demographics, disease location, tumor stage, treatment protocols, volumetric parameters, cut-off values determination method, cut-off values and follow-up. Data extraction discrepancies were resolved by consensus.

### 2.4. Study Quality Assessment

Two investigators (E.R. and A.D.V.) independently assessed the methodological quality of the included studies. The quality of each study was evaluated according to the National Institute for Health and Clinical Excellence (NICE) quality assessment tool [23] or the Consolidated Standards of Reporting Trials (CONSORT) [24], as appropriate. The revised CONSORT 2010 checklist contains 37 items (http://www.consort-statement.org; accessed on 12 February 2021), each of which was categorized as ‘yes’ if it is clearly and adequately reported, or ‘no’ if it is partially unclear or not reported at all. Compliance of more than 75% with CONSORT items was regarded as good quality, as previously performed in several studies [25,26].

### 2.5. Data Synthesis and Statistical Analysis

The data from each study were transcribed in table format. Dichotomous variables were reported by counts and percentages, while continuous variables were reported as median ± interquartile range (IQR: 25th and 75th).

The primary endpoints were overall survival (OS), disease-free survival (DFS) and locoregional control (LRC). OS was defined as the time from the date of diagnosis (or the start of treatment) to death by any cause. DFS was defined as the time from treatment until the recurrence of disease or death. LRC was defined as no evidence of tumor residual or (loco)regional recurrence during follow-up.

The impact of tumor volume on survival and disease control was measured by the effect size of the hazard ratio (HR). HRs and 95% confidence intervals (CIs) were extracted directly from each study if reported by the authors. Where both univariate and multivariate analyses were reported, HRs were extracted from multivariable models. Otherwise, they were estimated indirectly using the method described by Tierney, et al., [27]. Published Kaplan–Meier plots from each study were digitized using a GetData Graph Digitizer (version 2.26; http://getdata-graph-digitizer.com/index.php; accessed on 12 January 2021), and survival probabilities and follow-up times were extracted. Then, the number of subjects at risk, adjusted for censoring at different follow-up times, was calculated to reconstruct the HR estimate and its variance. An HR greater than 1 suggested a survival benefit for patients with low tumor volume, whereas an HR less than 1 suggested worse survival for patients with low tumor volume.

HRs were log-transformed before pooling effect size estimates. Cumulative Hazard Ratios with 95% CI are presented for the reported outcomes, calculated through the inverse variance method. Heterogeneity between studies was assessed by I^2^ statistics. Values of heterogeneity were evaluated according to the Cochrane manual of reviewers as follows: ≤40%, low heterogeneity; 41% to 60%, moderate heterogeneity; and 61% to 100%, substantial heterogeneity [28]. Given the observational nature of the majority of the included studies, we decided upon a statistically conservative approach. For this reason, we conducted our meta-analysis using a random-effects model, which ensures a more conservative estimate, assuming both within-study and between-study variability.

Meta-analyses were conducted by Review Manager (RevMan, version 5.3; The Nordic Cochrane Centre, The Cochrane Collaboration) and R software for statistical computing (R, version 3.4.0). Statistical significance was defined as *p* < 0.05.

## 3. Results

### 3.1. Search Results and Studies Description

The search strategy retrieved 1058 studies after duplicates removal. After screening the titles and abstracts, 1004 articles were rejected, while the remaining 54 papers were included for full text assessment. After applying the above-mentioned inclusion and exclusion criteria, a total of 10 [8,17,18,29,30,31,32,33,34,35] studies were included in our meta-analysis. Figure 1 shows both the reasons behind the exclusion of the non-eligible studies, and a flow chart of the study identification process. Full texts were considered not available if neither a specialized academic librarian nor the journal could provide them.

The general characteristics of the studies are shown in Table 1. A total number of 1417 patients with a median age of 59.3 years (IQR 57.5–60) were included. The median follow-up was 42 months (IQR 33–49). Detailed information about tumor location was reported by six studies as follows: base of tongue (*n* = 396, 43.14%), soft palate (*n* = 26, 2.83%), tonsillar fossa (*n* = 481, 52.4%) and posterior pharyngeal wall (*n* = 15, 1.63%). Three studies did not provide detailed information about tumor stages. The majority of the quantified patients were in a locally advanced stage (*n* = 601/714, 84.17%). HPV status was reported in three studies, with a total number of p16-positive patients of 268/468 (57.3%). The majority of patients (*n* = 808, 65.8%) were treated with concomitant chemoradiation therapy (CRT), while only 383 patients (31.2%) were treated with radiation therapy alone. Only three (0.25%) patients were treated with primary surgery alone, while another 33 (2.7%) patients received adjuvant (chemo)radiotherapy. Finally, 44 (3.6%) patients underwent salvage surgery. Tumor volume was calculated by the summation-of-areas technique (i.e., by multiplying the outlined area on every slice by the image reconstruction interval) in only two studies (*n* = 232, 16.4%), while in the others (*n* = 1185, 83.6%) it was calculated by an automatic system. Primary tumor volume (pTV), nodal tumor volume (nTV) and total tumor volume (tTV) were used in nine (*n* = 1230, 86.8%), five (*n* = 574, 40.5%) and three (*n* = 299, 21.1%) studies, respectively. Cut-off values were obtained through median values, mean values, log-rank test and receiver operating characteristic curve (AUC-ROC), with an average of 25.89, 20.79 and 38.34 cm^3^ for pTV, nTV and tTV, respectively.

### 3.2. Methodological Quality and Risk of Bias of Included Studies

The majority of the included studies were prospective [8,34] (*n* = 377, 26.6%) or retrospective [17,18,30,31,32,33,35] (*n* = 716; 50.5%) non-randomized studies. Otherwise, only a minority of patients (*n* = 324; 22.9%) were enrolled in a single randomized controlled trial (RCT) [29]. The results of the quality assessment of the included studies are reported in Table 2. All the included studies were of generally moderate/high quality, and satisfied at least five of the eight NICE quality assessment tool items. The main limitation was that the greatest number of cases (*n* = 716; 50.5%) were analyzed retrospectively [17,18,30,31,32,33,35]. However, all patients were recruited consecutively, regardless of study design. On the other hand, the only RCT included [29] showed a compliance of more than 75% with CONSORT items.

### 3.3. Overall Survival

The OS was analyzed using five studies with tumor volume (TV), reported as pTV by four studies and tTV by one study. tTV was assumed to be more influenced by primary tumor volume rather than by nodal volume, therefore tTV was comparable to pTV, and was included in the same analysis. In addition, one study provided survival analysis separately for p16-positive and p16-negative patients, and the two cohorts were analyzed independently. The pooled HR for death was 1.02 (*n* = 760; 95% CI, 0.99–1.05; *p* = 0.21). The test for heterogeneity gave significant results (χ^2^ = 22.19, *p* < 0.05, I^2^ = 77%). Two articles assessed the relation between nTV and OS. One study reported stratified data according to the HPV status. The pooled HR for death was 1.01 (*n* = 431; 95% CI, 1.00–1.02; *p* = 0.15). Heterogeneity was low, with a I^2^ of 0% (χ^2^ = 1.97, *p* = 0.37). Forest plots of HRs for OS with pTV and nTV are shown in Figure 2A,B, respectively.

### 3.4. Disease-Free Survival

DFS was analyzed using six studies with TV. When reported, tTV was included in the same analysis together with pTV. Data provided separately according to HPV status were analyzed independently. The pooled HR was 1.01 (*n* = 463; 95% CI, 1.00–1.03; *p* < 0.05). Heterogeneity was moderate, with a I^2^ of 53% (χ^2^ = 12.90, *p* < 0.05). Three articles analyzed the relation between nTV and DFS, with one of them providing survival analysis separately for p16-positive and p16-negative patients. The pooled HR was 1.02 (*n* = 175; 95% CI, 1.01–1.03; *p* < 0.05). There was no evidence of significant statistical heterogeneity (χ^2^ = 0.00, *p* = 1.00; I^2^ = 0%). Forest plots of HRs for DFS with pTV and nTV are shown in Figure 3A,B, respectively.

### 3.5. Locoregional Control

Five articles studied the relation between LRC and TV. One study reported data separately according to the primary site (tonsillar pillars/fossa vs. base of tongue vs. soft palate), while another study analyzed the effects of TV in two cohorts with different fractionation schedules (conventional vs. altered fractionation). Finally, another article provided data separately according to HPV status. In all cases, the different cohorts were added to the analysis independently. The pooled HR was 1.07 (95% CI, 0.99–1.17; *p* = 0.10). There was evidence of substantial heterogeneity, with a I^2^ of 68% (χ^2^ = 25.23, *p* < 0.05). Three articles analyzed the relation between nTV and LRC, and one of them provided data separately for p16-positive and p16-negative patients. The pooled HR was 1.02 (*n* = 122; 95% CI, 1.01–1.03; *p* < 0.05). The test for heterogeneity gave no significant results (χ^2^ = 0.66, *p* = 0.72; I^2^ = 0%). Forest plots of HRs for DFS with pTV and nTV are shown in Figure 4A,B, respectively.

## 4. Discussion

The current literature data are inconclusive about the prognostic role of tumor volume in OPSCC. Lack of high-quality RCTs and small-sample-sized studies prevent the drawing of any firm conclusion. To the best of our knowledge, this is the first meta-analysis to investigate the prognostic role of tumor volume in OPSCC. Our results showed that there was no significant correlation of pTV with locoregional failure and overall survival. Conversely, the high-pTV group was found to be associated with statistically significantly poorer DFS. However, the difference seems to be not clinically relevant, given the small effect size (HR 1.01). Nevertheless, heterogeneity between studies ranged from moderate to high: thus, these results should be interpreted cautiously. Moreover, the high-nTV group showed a significant difference in terms of DFS (HR 1.02) and LRC (HR 1.02), but not in OS (HR 1.01), when compared to the low-nTV group. Although nTV seemed to be a good prognostic factor in OPSCC, the advantage in terms of DFS and LRC was very small, raising some concerns about its prognostic role. In addition, the pooled data for nTV showed low heterogeneity, and this is probably related to the small number of studies included.

Several hypotheses have been postulated to explain why the tumor volume of OPSCC has not been shown to bear a significant correlation with survival and locoregional control rate. It has been suggested that infiltrating cancers tend to be less radiosensitive than exophytic cancers [36]. If compared with HNCs arising in other sites, OPSCCs show a large tendency for submucosal spread with varying degrees of infiltration. In addition, tumors of the tonsillar fossa seem to be more radiosensitive compared to base-of-tongue or soft palate tumors [37]. Thus, the different incidences of these subsets of OPSCC may confound the significance of tumor volume. Another explanation may be the different biological behavior and treatment response of HPV-related cancers. As shown in several studies [1,4,5,6], p16-positive tumors are generally smaller by T category compared with p16-negative tumors but have a greater tendency to spread to regional lymph nodes. Nevertheless, they are more (chemo)radiosensitive, thus resulting in higher rates of local control, which would again undermine the role of tumor volume. Finally, oropharyngeal tumors seem to be associated with a greater variation in tumor volume within a given stage, unlike other head and neck subsites [15,38]. For instance, Nathu, et al., [15] demonstrated a significant variation of oropharyngeal tumor volume for each T-stage, with a range of 0–32.5 cm^3^, 0–48 cm^3^ and 6.5–99.9 cm^3^ for T2, T3 and T4, respectively.

We recently raised some concerns regarding the adequacy of the current T staging system in OPSCCs [39]. Clinical T3 tumors are defined as lesions greater than 4 cm, or with extension to the lingual surface of the epiglottis. Although p16-positive tumors are mostly exophytic, the oropharynx is a very limited area, and a lesion greater than 4 cm is usually classified as a T4 tumor because of deep infiltration. Moreover, the definition of T4 tumors, based on extrinsic tongue muscles infiltration, does not seem adequate for all the subsets of OPSCCs. It is not rare to find a small carcinoma arising from the tonsillar fossa with minimal infiltration of the palatoglossus muscle. Such lesions are usually resectable with limited postoperative sequelae, thus making their classification as locally advanced tumors inappropriate. From this perspective, a specified TNM staging based on the oropharyngeal subsites rather than on a metric criterion may be valuable in the future.

This meta-analysis has several limitations, of which the main one is the relatively low quality of the included studies. Only one RCT was included [29], while the majority of papers were retrospective or prospective non-randomized studies [8,17,18,30,31,32,33,34,35]. Therefore, further RCTs should be conducted to better assess the prognostic role of tumor volume.

The second limitation of this study was that only a minority of papers provided information about HPV status. In particular, only one of the included studies [30] specifically investigated the prognostic value of primary and nodal tumor volume in p16-positive and p16-negative OPSCC patients, demonstrating that the first may be an independent prognostic factor in p16-negative patients, while the second may serve as an independent prognostic factor both in p16-positive and p-16 negative patients treated with ©RT. Thus, we did not determine the impact of HPV status on tumor volume. As mentioned above, HPV may be one confounding factor leading to the lower significance of tumor volume in OPSCCs, compared with other HNCs. Consequently, further studies are needed to investigate the relationship between tumor volume and HPV status.

Thirdly, although some authors found that patients with high pTV and nTV seem to be at higher risk of adverse events or death than patients with low pTV and nTV, they were unable to propose an optimal cut-off value to categorize the volumetric parameters as high or low. In previous studies, cut-off values for oropharyngeal cancer ranged from 22.8 and 35 cm^3^ [40,41]. Furthermore, Studer, et al., first ideated a volumetric staging system for HNC patients, defined by using three cut-off values (15/70/130 cm^3^), and demonstrated that it was most reliable for predicting outcome in a large cohort (*n* = 277) of definitively irradiated OPSCC patients [42]. Nevertheless, large study groups are needed to further confirm the reliability of such a volumetric staging system.

Moreover, sub-group analysis according to the type of treatment or other tumor-related prognostic factors could not be performed, given that stratified data were available only for a minority of patients.

Finally, although we tried to include only homogeneous studies in terms of volumetric parameters assessment, some difference existed among the included papers. Some studies used the summation-of-areas technique, while other studies used automatic volume measurements. Despite its accuracy, the summation-of-areas technique is time consuming and may be encumbered with a certain degree of interobserver variability. Therefore, a consensus should be reached on standardization of volume measurements to further allow homogeneous volume measurements to be adequately compared.

## 5. Conclusions

The results of this analysis suggested that pTV and nTV are not predictors of OS, and that they may not be used as prognostic factors in OPSCCs. Moreover, the difference in terms of DFS and LRC was too small to appear clinically relevant. Further RCTs should be conducted to assess the prognostic role of tumor volume stratifying HPV-positive and HPV-negative OPSCCs. In addition, prospective and multicenter studies using a standardized volume measurement method are still needed to provide generalizable cut-off values.

## Figures and Tables

**Figure 1 cancers-14-02465-f001:**
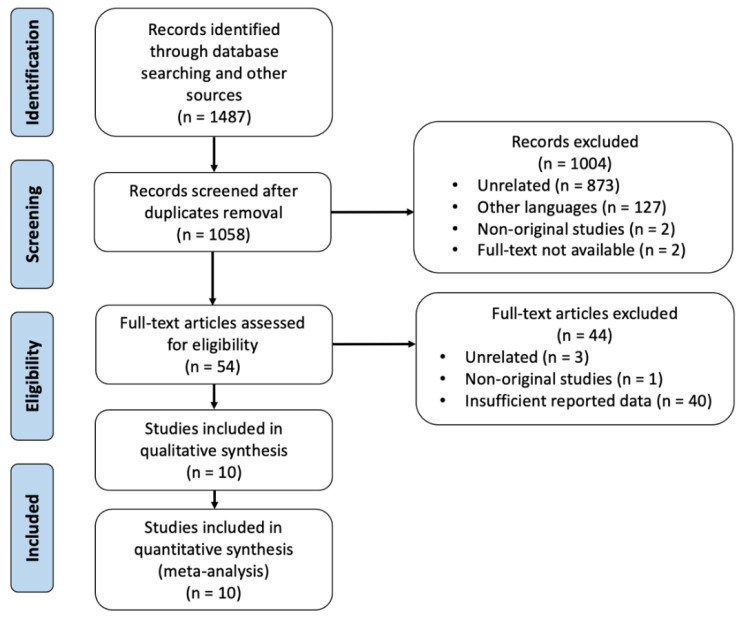
PRISMA 2009 flow diagram.

**Figure 2 cancers-14-02465-f002:**
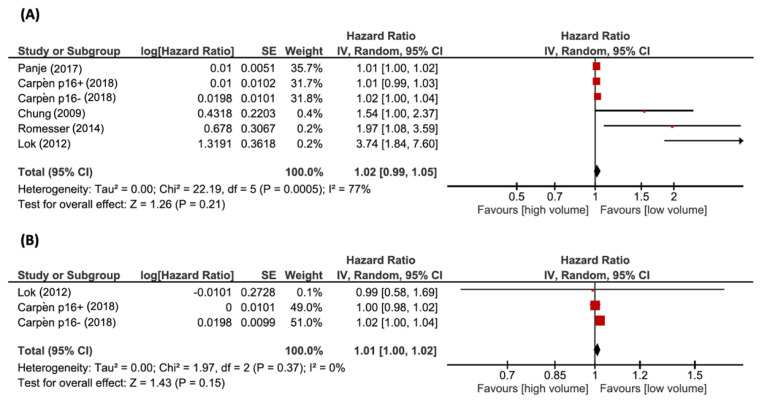
Forest plots of HRs for OS with (**A**) pTV and (**B**) nTV. References: Panje et al. (2017) [34]; Carpèn et al. (2018) [30]; Chung et al. (2009) [31]; Romesser et al. (2014) [35]; Lok et al. (2012) [18].

**Figure 3 cancers-14-02465-f003:**
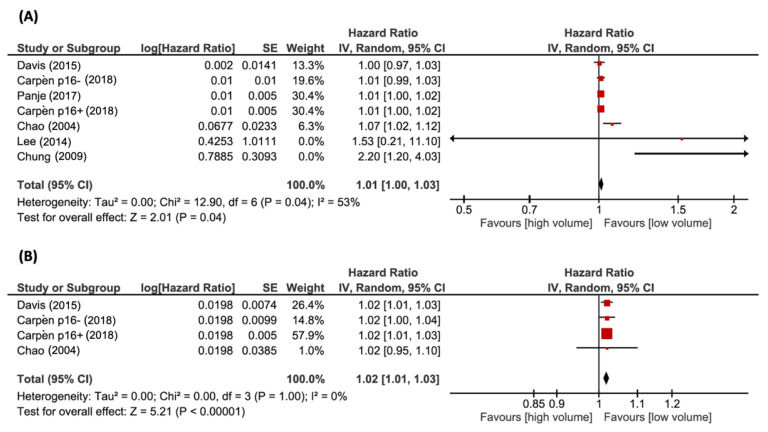
Forest plots of HRs for DFS with (**A**) pTV and (**B**) nTV. References: Davis et al. (2015) [32]; Carpèn et al. (2018) [30]; Panje et al. (2017) [34]; Chao et al. (2004) [17]; Lee et al. (2014) [33]; Chung et al. (2009) [31].

**Figure 4 cancers-14-02465-f004:**
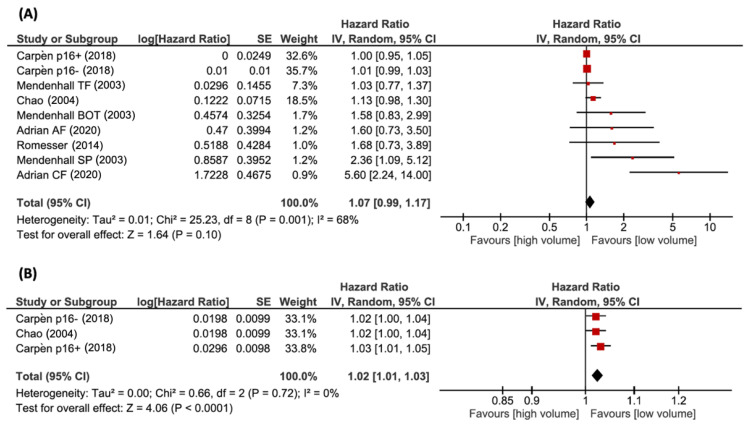
Forest plots of HRs for LRC with pTV (**A**) and nTV (**B**). References: Carpèn et al. (2018) [30]; Mendenhall et al. (2003) [8]; Chao et al. (2004) [17]; Adrian et al. (2020) [29]; Romesser et al. (2014) [35].

**Table 1 cancers-14-02465-t001:** General characteristics of the studies, and oncologic outcomes.

Study	Study Design	No. (Male)	Age (Range)	No. p16+	FU (m) (Range)	Stage	T Site	Tx	Mean pTV (cm^3^) (Range)	Mean nTV (cm^3^) (Range)	Mean tTV (cm^3^) (Range)	Cut-Off Type	Cut-Off pTV (cm^3^)	Cut-Off nTV (cm^3^)	Cut-Off tTV (cm^3^)
[29] **CF**	RCT	160 (121)	58 (35–86)	69	60 (N/A)	N/A	N/A	RT (*n* = 160)	23.7 (20.2–27.17)	N/A	N/A	N/A	23	N/A	N/A
[29] **AF**	RCT	164 (118)	59 (32–80)	74	60 (N/A)	N/A	N/A	RT (*n* = 164)	25.3 (21.3–29.28)	N/A	N/A	N/A	23	N/A	N/A
[30] **p16+**	R	72 (58)	61 (41.4–84.7)	72	31 (N/A)	I (*n* = 39) II (*n* = 15) III (*n* = 18) IV (*n* = 0)	N/A	N/A	23 (16.3–29.7)	26 (18.1–33.88)	N/A	Mean	23	26	N/A
[30] **p16−**	R	19 (14)	66 (54.9–81.7)	0	31 (N/A)	I (*n* = 2) II (*n* = 7) III (*n* = 0) IV (*n* = 10)	N/A	N/A	38 (17.4–58.62)	13 (0–28.38)	N/A	Mean	38	13	N/A
[17]	R	31 (N/A)	N/A	N/A	33	N/A	N/A	RT (*n* = 14) CRT (*n* = 17)	30.5 (22.7–38.3)	23.2 (16–30.4)	N/A	N/A	N/A	N/A	N/A
[31]	R	42 (29)	59.6 (28–85)	N/A	33.3 (4–105)	I (*n* = 2) II (*n* = 6) III (*n* = 8) IV (*n* = 26)	BOT (*n* = 13) SP (*n* = 3) TF (*n* = 26) PPW (*n* = 0)	CRT (*n* = 6) PS (*n* = 3) PS+(C)RT (*n* = 33)	N/A	N/A	N/A	Log-rank test	3.5	N/A	N/A
[32]	R	53 (50)	55.4 (39–81)	53	29 (4–76)	N/A	N/A	RT (*n* = 1) CRT (*n* = 52)	18.8 (13.1–24.46)	26.75 (16.5–37.01)	44.49 (32.37–56.61)	N/A	N/A	N/A	N/A
[33]	R	59 (48)	60 (43–86)	N/A	41.3 (9.3–73.5)	I (*n* = 2) II (*n* = 7) III (*n* = 12) IV (*n* = 38)	BOT (*n* = 12) SP (*n* = 0) TF (*n* = 44) PPW (*n* = 3)	RT (*n* = 9) CRT (*n* = 50)	13.79 (1.1–50.81)	16.75 (0–140.07)	30.54 (1.1–148.9)	Log-rank test	15	25	35
[18]	R	340 (293)	N/A	N/A	34 (5–67)	I (*n* = 3) II (*n* = 12) III (*n* = 62) IV (*n* = 263)	BOT (*n* = 162) SP (*n* = 4) TF (*n* = 166) PPW (*n* = 8)	RT (*n* = 17) CRT (*n* = 323) SS (*n* = 44)	42.53 (4.1–306.6) *	19.04 (0–442.05) *	N/A	Median	32.79	19.04	N/A
[8]	P	190 (N/A)	N/A	N/A	42 (2–241)	N/A	BOT (*n* = 72) SP (*n* = 12) TF (*n* = 106) PPW (*n* = 0)	CRT (*n* = 190)	BOT: 24.4 (1.5–235)	N/A	N/A	Median	BOT: 14.75	N/A	N/A
									SP: 11.8 (0–99.9)				SP: 5.2		
									TF: 18.2 (0–187.5)				TF: 9.2		
[34]	P	187 (134)	61.6 (36.9–91.4)	N/A	61.2 (1.7–169)	I (*n* = 0) II (*n* = 15) III (*n* = 35) IV (*n* = 129) Rec (*n* = 3)	BOT (*n* = 84) SP (*n* = 4) TF (*n* = 97) PPW (*n* = 2)	RT (*n* = 7) CRT (*n* = 180)	N/A	N/A	40 (3–216) *	N/A	N/A	N/A	15/70/130
[35]	R	100 (86)	56 (27–81)	N/A	49 (N/A)	N/A	BOT (*n* = 53) SP (*n* = 3) TF (*n* = 42) PPW (*n* = 2)	CRT (*n* = 100)	40.7 (N/A) *	N/A	N/A	Median	40.7	N/A	N/A

* Median value with interquartile range (IQR). Abbreviations: RCT—randomized controlled trial, P—prospective cohort, R—retrospective cohort, FU—follow-up, Tx—treatment; pTV—primary tumor volume, nTV—nodal tumor volume, tTV—total tumor volume, CF—conventional fractionation, AF—altered fractionation, BOT—base of tongue, SP—soft palate, TF—tonsillar fossa, PPW—posterior pharyngeal wall, RT—radiotherapy, CRT—chemoradiotherapy, PS—primary surgery, SS—salvage surgery.

**Table 2 cancers-14-02465-t002:** Quality Assessment of case series studies checklist from National Institute for Health and Clinical Excellence.

Study	Multicenter	Aim	Inclusion/Exclusion Criteria	Outcome	Prospective	Consecutive	Main Findings	Stratified
[30]	No	Yes	Yes	Yes	No	Yes	Yes	Yes
[17]	No	Yes	No	Yes	No	Yes	Yes	Yes
[31]	No	Yes	Yes	Yes	No	Yes	Yes	Yes
[32]	No	Yes	Yes	Yes	No	Yes	Yes	Yes
[33]	No	Yes	Yes	Yes	No	Yes	Yes	Yes
[18]	No	Yes	Yes	Yes	No	Yes	Yes	Yes
[8]	No	Yes	No	Yes	Yes	Yes	Yes	Yes
[34]	No	Yes	No	Yes	Yes	Yes	Yes	Yes
[35]	No	Yes	Yes	Yes	No	Yes	Yes	Yes

## Data Availability

The data presented in this study are available on request from the corresponding author.

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
