# Peer review of "Does Tumor Volume Have a Prognostic Role in Oropharyngeal Squamous Cell Carcinoma? A Systematic Review and Meta-Analysis"

_cancers, 2022, doi:10.3390/cancers14102465_

Round 1
Reviewer 1 Report
This meta-analysis is nicely done, although limited by lack of HPV status. A statement summarizing what the one analysis that stratified by HPV status did find would be useful for context, in intro or discussion. Also, year of publication is somewhat of a surrogate for HPV status in that the earlier manuscripts probably had fewer HPV+ patients. I think it would be helpful to the reader to include year of publication in the forest plots next to the author names.
Reviewer 2 Report
-
Figure 1 shows that 1004 records are excluded due to different reasons. Since you have mentioned these categories of exclusion, it would be interesting to specify the number of records for each category. Please specify what actions have been taken to determine that a full-text is not available; is it a question of the year of publication? or perhaps subscriptions of your institutions?. Depending on the criteria used, this could mean an ascertainment bias that should perhaps be taken into account. - Please specify on what date the search has been carried out.
- In Table 1 there are two studies that present the results divided into two row groups (Adrian et al divide into conventional vs. altered fractionation and Carpén et al according to p16); Although this is already commented on in some section of the results, it should also be clarified in the table.
- The years of publication specified in Table 2 should be reviewed as they do not match. In addition, I suggest adding et al. in each row.
- The biases explained by the authors are already well explained and are important enough (they do not know in many cases the p16 and that studies are very retrospective). It would add to the limitations of the study:
- -No treatment or other survival factors were taken into account.
- - No multivariate. They include patients operated on, with adjuvant, treated only with RT or RT + QT but it is not explained.
- -which is difficult to know if there is homogeneity in knowing how the PTV is defined
